# Diabetes Mellitus in Pancreatic Cancer: A Distinct Approach to Older Subjects with New-Onset Diabetes Mellitus

**DOI:** 10.3390/cancers15143669

**Published:** 2023-07-19

**Authors:** Jan Bures, Darina Kohoutova, Jan Skrha, Bohus Bunganic, Ondrej Ngo, Stepan Suchanek, Pavel Skrha, Miroslav Zavoral

**Affiliations:** 1Institute of Gastrointestinal Oncology, Military University Hospital Prague, 169 02 Prague, Czech Republic; stepan.suchanek@uvn.cz (S.S.); miroslav.zavoral@uvn.cz (M.Z.); 2Department of Medicine, First Faculty of Medicine, Charles University, Prague and Military University Hospital Prague, 169 02 Prague, Czech Republic; bohus.bunganic@uvn.cz; 3Biomedical Research Centre, University Hospital Hradec Kralove, 500 03 Hradec Kralove, Czech Republic; darina.kohoutova@rmh.nhs.uk; 4The Royal Marsden NHS Foundation Trust, London SW3 6JJ, UK; 5Third Department of Internal Medicine—Endocrinology and Metabolism, First Faculty of Medicine, Charles University, Prague and General University Hospital in Prague, 128 08 Prague, Czech Republic; jan.skrha@lf1.cuni.cz; 6Institute of Health Information and Statistics of the Czech Republic, 128 01 Prague, Czech Republic; ondrej.ngo@uzis.cz; 7Institute of Biostatistics and Analyses, Faculty of Medicine, Masaryk University, 602 00 Brno, Czech Republic; 8Department of Medicine, Third Faculty of Medicine, Charles University, Prague and University Hospital Kralovske Vinohrady, 100 00 Prague, Czech Republic; pavel.skrha@fnkv.cz

**Keywords:** age over 60 years, diagnostic algorithm, new-onset diabetes mellitus, pancreatic ductal adenocarcinoma

## Abstract

**Simple Summary:**

Pancreatic ductal adenocarcinoma remains one of the most serious malignancies and a leading cause of cancer-related deaths worldwide. There are no effective screening methods available so far, even for high-risk individuals. At the time of diagnosis, impaired glucose metabolism is present in about 3/4 of all patients. Several types of diabetes mellitus can be found in pancreatic cancer; however, type 2, pancreatic-cancer-associated type 3c, and diabetes mellitus associated with non-malignant diseases of the exocrine pancreas (with a reduction or loss of islet-cell mass) are the most frequent ones. This paper proposed a distinct approach to older subjects with new-onset diabetes mellitus with possible pancreatic cancer. It could improve the current unsatisfactory situation in diagnostics and subsequent poor outcomes of treatment of pancreatic ductal adenocarcinoma.

**Abstract:**

Background: Pancreatic ductal adenocarcinoma (PDAC) is associated with a very poor prognosis, with near-identical incidence and mortality. According to the World Health Organization Globocan Database, the estimated number of new cases worldwide will rise by 70% between 2020 and 2040. There are no effective screening methods available so far, even for high-risk individuals. The prognosis of PDAC, even at its early stages, is still mostly unsatisfactory. Impaired glucose metabolism is present in about 3/4 of PDAC cases. Methods: Available literature on pancreatic cancer and diabetes mellitus was reviewed using a PubMed database. Data from a national oncology registry (on PDAC) and information from a registry of healthcare providers (on diabetes mellitus and a number of abdominal ultrasound investigations) were obtained. Results: New-onset diabetes mellitus in subjects older than 60 years should be an incentive for a prompt and detailed investigation to exclude PDAC. Type 2 diabetes mellitus, diabetes mellitus associated with chronic non-malignant diseases of the exocrine pancreas, and PDAC-associated type 3c diabetes mellitus are the most frequent types. Proper differentiation of particular types of new-onset diabetes mellitus is a starting point for a population-based program. An algorithm for subsequent steps of the workup was proposed. Conclusions: The structured, well-differentiated, and elaborately designed approach to the elderly with a new onset of diabetes mellitus could improve the current situation in diagnostics and subsequent poor outcomes of therapy of PDAC.

## 1. Introduction

Pancreatic cancer remains one of the most serious malignancies and a leading cause of cancer-related deaths worldwide [1]. According to the GLOBOCAN database, there were nearly 500 thousand new cases of pancreatic cancer worldwide in 2020 and about 470 thousand pancreatic cancer-related deaths. According to the World Health Organization, the estimated number of new cases worldwide is predicted to rise by 70% between 2020 and 2040 [2,3,4]. Pancreatic ductal adenocarcinoma (PDAC) is the most common histological type of malignant pancreatic tumor (>90%) [5]. However, there are no guidelines on population-based screening for PDAC. Its high lethality (three-year survival less than 10%) and mortality rate (almost the same as prevalence) are attributed to pancreatic cancer biology (including the early development of metastasis and poor response to systemic treatments), anatomy (close relationship to large vessels in the retroperitoneum), and difficulties with making a timely diagnosis combined with a lack of standardized international guidelines for the evaluation of suspicious pancreatic lesions (including cysts) [6,7].

At the time of PDAC diagnosis, impaired glucose metabolism is present in about 3/4 of all patients [8,9,10,11,12]. In a large prospective study (512 newly diagnosed PDAC individuals and 933 controls), 86% of PDAC subjects had either impaired fasting glucose (38.5%) or diabetes mellitus (47.5%) [8]. According to a PubMed search (https://pubmed.ncbi.nlm.nih.gov) (accessed on 9 April 2023), there are nearly 4200 records on “pancreatic cancer + diabetes mellitus”. Our aim was not to provide a comprehensive literature review or to try to create a meta-analysis based on the very heterogeneous studies available so far. We focused instead on providing an explanation of the different types of diabetes mellitus associated with pancreatic cancer and outlining a possible future basis for PDAC screening and/or diagnosis at the early stage of the disease.

First of all, it is important to briefly review the current terminology, with particular emphasis on the difference between screening, early diagnosis, and diagnosis at an early stage of PDAC.

## 2. Screening for PDAC in the General Population

Screening refers to the use of simple tests across a healthy population to identify those individuals who have a disease but have been asymptomatic in relation to that particular disease so far. Based on the existing evidence, average-risk population screening can only be advocated for cervical, breast, and colorectal cancer [13].

Many different methods and possible markers for PDAC screening have been proposed, including microRNA and other non-coding RNAs, cell-free DNA (cfDNA), circulating tumor DNA (ctDNA), lipidomic profiling and other metabolomics, combined spectroscopy analyses (electron circular dichroism, Raman optical activity, infrared spectroscopy), exosomes, S100 proteins, CEMIP (cell migration-inducing hyaluronan binding protein) and other proteomics, different enzymatic activities (e.g., dipeptidyl peptidase-IV or M2-pyruvate kinase), or multiple metabolites (e.g., dipeptides glycylvaline, aspartylphenylalanine, pyroglutamylglycine, phenylalanylphenylalanine, phenylalanylleucine, tryptophylglutamate) [14,15,16,17,18,19,20,21,22,23,24,25,26,27,28,29,30,31,32,33,34,35,36,37,38,39,40,41,42,43,44,45,46,47,48,49,50,51,52,53,54]. Several panels of combined possible biomarkers have been proposed, starting from 58 different investigations of sera [55], multi-analyte blood tests—a CancerSEEK panel [56], polygenic and multifactorial scores (PANcreatic Disease ReseArch—PANDoRA consortium) [57], and more than 300 candidate biomarkers in different media (serum, urine, saliva, pancreatic juice, cyst fluid analysis, feces) [Reference [58]; reviewed by O’Neill et al. in detail, with 456 references]. Several clinical risk prediction models for pancreatic cancer have been developed for different target populations [59,60], and machine learning-based models for the detection of PDAC have been developed [61]. However, the costs and/or limited accuracy of the above-mentioned methods, along with the current relatively low incidence of PDAC worldwide (despite its fatal course), remain the main reasons why no reliable and/or feasible screening programs for PDAC have been introduced into clinical practice in the general population so far [62,63].

## 3. Screening for PDAC in High-Risk Subjects

Average-risk sporadic PDAC is responsible for about 90% of all cases of pancreatic cancer. The rest comprises high-risk subjects with a family history of PDAC (at least two first-degree relatives with PDAC), underlying inherited diseases (e.g., Peutz–Jeghers syndrome, hereditary pancreatitis, Lynch syndrome) and/or patients with known genetic mutations investigated due to another indication (e.g., mutations in CDKN2A, BRCA1, BRCA2, PALB2, and ATM genes) [64,65,66,67].

A unique prospective long-term follow-up study enrolled 354 individuals at high risk for PDAC (based on genetic factors or family history). Overall, 24 of 354 patients (7%) had neoplastic progression (14 PDAC and 10 high-grade dysplasia) over a 16-year period. Magnetic resonance imaging (MRI) and endoscopic ultrasound (EUS) were the two major methods used for a follow-up [48,64,65,66,68].

The Japanese Pancreas Society, AGA (American Gastroenterological Association), and ASGE (American Society for Gastrointestinal Endoscopy) have recently published guidelines on screening and early diagnosis of PDAC in high-risk subjects [66,69,70,71]. The Italian cooperative group has prepared a multidisciplinary consensus statement on the early prediction of pancreatic cancer from new-onset diabetes mellitus [72]. The Italian Association of Medical Oncology, Italian Medical Diabetologists Association, Italian Society of Endocrinology, and Italian Society of Pharmacology addressed the feasibility of screening for early PDAC in patients with diabetes mellitus, reviewed a set of update statements and suggested a decision aid tool for daily clinical practice [72].

## 4. Surveillance of Pancreatic Cystic Lesions

European evidence-based guidelines are available for the diagnosis and management of pancreatic cystic neoplasms [73]. Mucinous cysts, including intraductal papillary mucinous neoplasms (IPMN) and mucinous cystic neoplasms (MCN), are the two main precancerous conditions that require follow-up and surgical intervention if indicated [73,74]. MRI and EUS (with fine-needle aspiration/biopsy) are the most suitable methods of follow-up for these patients. Relative indications for surgery in IPMN include a main pancreatic duct diameter between 5 and 10 mm, cyst diameter of ≥40 mm, new onset of diabetes mellitus, an enhancing nodule less than 5 mm in size, grow-rate ≥ 5 mm/year, increased serum CA 19-9 concentration (≥37 U/mL) and/or acute pancreatitis (caused by IPMN) [73]. Absolute indications for surgery in IPMN, due to the high risk of malignant transformation, include positive cytology for high-grade dysplasia, solid mass, tumor-related jaundice, an enhancing mural nodule ≥ 5 mm, and a main pancreatic duct diameter > 10 mm [73]. Cyst fluid analysis may also be helpful as high CEA and low glucose concentration might suggest a higher risk of progression into cancer and are associated with the mucinous character of the cyst [75,76,77,78,79]. Based on a meta-analysis (11 studies; nearly 4 thousand patients), risk factors for the malignant transformation of pancreatic cystic neoplasms include diabetes mellitus, older age, and male gender [80]. In a large study (137,970 patients with a pancreatic cyst, of which 14,279 had a new diagnosis; median follow-up 42 months), the risk of progression into PDAC was higher in those with new-onset diabetes (hazard ratio 2.8) compared to those with preexisting long-term diabetes mellitus (hazard ratio 1.59) [81]. The mean interval between new-onset diabetes and cancer diagnosis was 7.5 months [81].

## 5. Diagnosis of PDAC in Its Early Stage

The preferred staging system used for all pancreatic cancers is the TNM classification (tumor, node, metastasis) of the combined American Joint Committee on Cancer and the Union for International Cancer Control [82]. High-grade pancreatic intraepithelial neoplasia (PanIN-3) is a premalignant lesion classified as carcinoma in situ [65]. According to a multi-institutional validation study by the American Joint Commission on Cancer (with 2318 PDAC individuals), statistically appropriate prognostic cut-offs for tumor size were defined as <2.2 cm and >4.8 cm [83]. However, it is debatable whether the T1B stage (a tumor 2–4 cm in its greatest dimension) could still be considered an early stage of PDAC. The prognosis of PDAC, even at its early stage, is still very unsatisfactory.

## 6. Early Diagnosis of Pancreatic Cancer

Early detection does not necessarily mean diagnosis of early-stage PDAC. Early diagnosis programs aim at reducing the proportion of patients who are diagnosed at a late stage. They have three main components: increased awareness of the first signs of cancer (among physicians, nurses, and other health care providers as well as among the general public); improved accessibility and affordability of diagnosis and treatment services, and improved referral from the first to secondary and tertiary levels of care [13]. Unique French cooperative guidelines tackle the early detection of PDAC briefly [84].

## 7. Diabetes Mellitus and Pancreatic Cancer

Great attention has been paid to the relationship between diabetes mellitus and pancreatic cancer. Many large studies have addressed this issue. A Nurses’ Health Study and a Health Professionals Follow-Up Study (with repeated assessments over 30 years) enrolled 112,818 women and 46,207 men. In total, 1116 incident cases of pancreatic cancers were identified. New-onset diabetes mellitus (HR 2.97) accompanied by weight loss (>4 kg; HR 6.75) was associated with a substantially increased risk of developing pancreatic cancer. Older age, previous “healthy weight,” and no intentional weight loss elevated this risk further [35]. Diabetes mellitus is more prevalent (47% vs. 7%) in PDAC compared with controls [8]. New-onset diabetes (<2-year duration), older age, higher body-mass index (before the illness), and greater frequency of family history of diabetes mellitus were further discriminating features [85,86].

Major shortcomings of a substantial amount of studies are caused by the fact that they do not distinguish (define) particular types of diabetes mellitus, mainly type 2, from others. Notably, “pancreatogenic diabetes mellitus” [87,88,89] is a deceptive term as it comprises a diverse set of aetiologies, including PDAC-associated diabetes mellitus and diabetes mellitus associated with chronic non-malignant diseases of the exocrine pancreas. Hard data on the prevalence of “pancreatogenic diabetes” is scarce due to a lack of research on this issue and the difficulties encountered in classification, possibly causing misdiagnosis and underestimation [90]. Some authors estimate that PDAC-associated type 3c diabetes mellitus is present in about 1% of individuals with new-onset diabetes over 50 [1]. However, the real prevalence is likely to be substantially higher. Nearly one-third of new-onset diabetes mellitus in pancreatic cancer remains undiagnosed [91]. Based on the Czech Registries [92,93], coincident PDAC and new-onset diabetes mellitus account for 5.3% of all newly diagnosed diabetes mellitus of any type in subjects over 60 years of age. Another issue that may prevent understanding on a pathophysiological basis is the fact that previous long-standing type 2 or diabetes mellitus associated with chronic non-malignant diseases of the exocrine pancreas can be combined with a new onset of PDAC-associated type 3c diabetes mellitus when the cancer develops and thus substantially modifying the clinical features.

## 8. Different Types of Diabetes Mellitus in Pancreatic Cancer

According to the American Diabetes Association (ADA), diabetes mellitus is classified into four major groups: type 1, type 2, specific types, and gestational diabetes mellitus [94]. Apart from other subgroups, “specific types” also include diabetes mellitus associated with different diseases of the exocrine pancreas [94]. “Pancreatic diabetes” is a proposed “umbrella term” despite the broad range of different etiology and pathogenesis, distinct clinical features, and prognosis. Diabetes mellitus in the context of diseases of the exocrine pancreas has also been termed “pancreoprivic diabetes” [95]. Hyperglycemia due to general pancreatic dysfunction has been called “type 3c diabetes mellitus” [94]. The WHO classification of diabetes mellitus uses the term “diabetes due to diseases of the exocrine pancreas” rather than type 3c diabetes mellitus [96]. In contrast to the current ADA classification of “pancreatic diabetes mellitus” [94], another classification should be introduced taking into account the different etiology, pathogenesis, and clinical features of particular types of diabetes mellitus in pancreatic diseases.

Regarding a practical clinical approach, the current pathophysiology basis should be maintained as it is determinative for both diagnosis and therapy. Thus several types of diabetes mellitus can be found in PDAC; however, type 2, PDAC-associated type 3c, and diabetes mellitus associated with non-malignant diseases of the exocrine pancreas (with a reduction or loss of islet-cell mass) are more frequent [87,97] (see Table 1 for basic characteristics).

### 8.1. Type 2 Diabetes Mellitus

Type 2 diabetes mellitus is typically a part of metabolic syndrome (a group of conditions including arterial hypertension, obesity, and atherogenic dyslipidemia). It is characterized by insulin resistance as the first event, resulting in hyperinsulinemia [90,94]. Long-standing type 2 diabetes mellitus is a risk factor for the development of PDAC (a risk increase of 10%, according to a meta-analysis of 151 cohorts with >32 million people) [112].

Based on a meta-analysis, therapy of type 2 diabetes using metformin could reduce the risk of PDAC by 18% compared with other treatments [113]. However, there is no full agreement on this issue. A Korean study found that women with type 2 diabetes that use metformin are at a higher risk of PDAC than women with diabetes mellitus that do not [114].

### 8.2. PDAC-Associated Type 3c Diabetes Mellitus

PDAC-associated type 3c diabetes mellitus has a typical onset in older subjects (over 60 years of age) [101]. This type of diabetes mellitus is prevalent even in small pancreatic cancers and occurs before the PDAC is radiologically detectable. There are some features resembling the paraneoplastic phenomenon of PDAC-associated type 3c diabetes mellitus [99,101]. That is why some authors suggest that PDAC-associated type 3c diabetes mellitus is a paraneoplastic syndrome [99,101,112,113,114,115,116,117,118,119,120]. In this type of diabetes, weight loss often precedes the onset of pancreatic cancer symptoms by several months. This initial period of weight loss cannot be attributed to cachexia. Weight loss, associated with diabetes and occurring prior to the onset of cachexia, is considered to be a paraneoplastic phenomenon induced by pancreatic cancer [99].

PDAC-associated type 3c diabetes is characterized by a decreased secretion of insulin and increased insulin resistance [99,101]. This type of diabetes mellitus is harder to control. Its instability is explained by the variable need for insulin therapy, together with insulin resistance and rapid weight loss [121].

The pathogenesis of pancreatic cancer-associated diabetes has not been fully elucidated yet; see References [99,101] for a detailed review. Supernatants from pancreatic cell lines have been shown to induce both insulin resistance and beta-cell dysfunction. There is a presumption that this paraneoplastic beta-cell dysfunction is mediated by exosomes carrying content toxic to beta-cells [115]. Adrenomedullin receptor blockades remove the inhibitory effect of exosomes on insulin secretion. Beta-cells exposed to adrenomedullin or pancreatic cancer exosomes show upregulation of endoplasmic reticulum stress genes and increased reactive oxygen/nitrogen species. Adrenomedullin binds to its receptor on the surface of beta-cells in order to inhibit insulin secretion [115].

Insulin resistance in PDAC is presumed to occur at the postreceptor level. In searching for the exact mechanism, islet amyloid polypeptide (IAPP) was proposed as a putative mediator. The serum level of IAPP was higher in pancreatic cancer patients than in the diabetic population or healthy controls [122]. IAPP is also known to cause insulin resistance in skeletal muscles. Normally, beta-cells secrete IAPP along with insulin. However, pancreatic cancer induces beta-cells to selectively secrete IAPP [122].

One-third to one-half of type 3c diabetes mellitus might be resolved, or at least improved, after successful surgical resection of PDAC, despite the reduction of islet-cell mass resulting from the surgery [123]. A study of 41 diabetic PDAC individuals who underwent pancreaticoduodenectomy confirmed that diabetes mellitus resolved postoperatively in 57% of those with new-onset diabetes, and none of the patients with long-standing diabetes experienced any improvement in glycemic status [124]. These findings further support the paraneoplastic basis of PDAC-associated type 3c diabetes mellitus [99,101].

### 8.3. Diabetes Mellitus Associated with Chronic Non-Malignant Diseases of the Exocrine Pancreas

This type of diabetes mellitus is characterized by decreased insulin secretion and normal insulin sensitivity. It is a secondary complication of different advanced chronic non-malignant diseases of the pancreas, usually preceded or associated with the exocrine pancreatic insufficiency, with a reduced or lost islet-cell mass (e.g., in chronic alcoholic pancreatitis, cystic fibrosis, or as a consequence of pancreatectomy) [94]. Alone, this type of diabetes mellitus has no role in the pathogenesis of pancreatic cancer. However, underlying disease can be associated with a higher risk of PDAC development (e.g., chronic alcoholic pancreatitis by 4% after 15 to 20 years) [125].

## 9. Diagnosis

We strive to propose a set of investigations that would be feasible, accessible, and affordable on a broad population basis (not just for research purposes). The diagnosis of PDAC-associated type 3c diabetes, when the pancreatic disease is not overt, can be challenging. Therefore, new-onset diabetes in subjects over the age of 60 and associated with weight loss should prompt consideration of this type of diabetes. Rapid progression of hyperglycemia or an early need for insulin should also prompt to consider this diagnosis [126].

Diagnosis of PDAC-associated type 3c diabetes mellitus is based on a decrease in insulin production (serum insulin, C-peptide) and an increase in insulin resistance (fasting glycemia, serum insulin and insulin resistance indices—HOMA-IR or QUICKI), see Table 1.

Several insulin resistance indices were proposed [127,128,129,130,131,132,133,134]. Although they are not commonly used in routine clinical practices, they are easily calculated. The most frequently validated indices are HOMA and QUICKI [135,136,137,138]. HOMA and QUICKI indices are robust, accurate, and reproducible methods that appropriately predict changes in insulin sensitivity after therapeutic interventions as well as the onset of diabetes [127,128,129,130,131,132,133,134,135,136,137,138]. They correlate with euglycemic insulin clamps [139,140,141]. Insulin resistance, assessed by HOMA and QUICKI indices, improved after pancreatoduodenectomy in patients with pancreatic cancer [142]. HOMA-IR (HOmeostatic Model A assessment for Insulin Resistance) is calculated as follows: serum fasting insulin × glycemia/22.5 (for glucose concentration in mmol/L) or serum fasting insulin × glycemia/405 (for glycemia in mg/dL; in both cases, insulin concentration is in mU/mL). Internet calculators are available: https://www.omnicalculator.com/health/homa-ir, (accessed on 27 June 2023) The normal value of HOMA-IR is <2.5 [114]. QUICKI (QUantitative Insulin sensitivity ChecK Index) is calculated as follows: 1/[log(I_0_) + log(G_0_)], where I_0_ is the fasting insulin (mU/mL), and G_0_ is the fasting glucose concentration (mg/dL). The normal value of QUICKI is >0.45 [128].

An increased glucagon/insulin ratio is another marker of insulin resistance [109]. Investigation of glucagon has not been extensively used (due to a demanding preanalytical phase). Euglycemic insulin clamps are not feasible in routine clinical practice.

## 10. Algorithm of the Management of New-Onset Diabetes Mellitus of Any Type in Older Subjects in Relation to PDAC

At present, there is no feasible, effective, and reliable population-based program for the diagnostics of PDAC. Continuously rising incidence and ongoing poor outcomes in PDAC treatment mean that a new strategic approach is required urgently. A program based on a distinct attitude to subjects with new-onset diabetes mellitus of different types might be one way, at least according to data from the Czech Republic (although this does not deal with younger patients and/or subjects without diabetes mellitus). The incidence of pancreatic cancer in the Czech Republic is the 3rd highest in Europe, with 2332 (2018) and 2466 (2020) new cases per year (i.e., incidence 23.0/100 thousand per year). The median age at diagnosis was 70 (interquartile range 63–78) [92]. Incidence of newly diagnosed diabetes mellitus of any type requiring pharmacotherapy, has been relatively stable in the Czech Republic within recent years, starting from 64,486 patients (2015), through 60,718 (2017) and 57,331 (2020) to 70,895 (2021) new cases per year (mean rate 62,471 new cases; mean incidence 5950/100 thousand general population per year). About one-half of new cases have been diagnosed among people over 60 (males to females 53.3:46.7%) [93]. An algorithm for the distinct management of subjects over 60 with new-onset diabetes mellitus of any type (diagnosed within the last 24 months) is proposed in Figure 1.

Less than 20% of newly diagnosed PDAC patients undergo surgical resection with curative intent [143,144]. About 50% of PDAC cases are diagnosed as advanced ones (pancreatic mass ≥ 4 cm) and/or metastatic disease [83].

New-onset diabetes mellitus, together with significant weight loss, serves as an important early discriminating factor (Figure 1). A fully individualized approach is mandatory as an example of indispensable personalized medicine, even upon a general-population basis. Patients showing diabetes mellitus with decreased insulin secretion and increased insulin resistance must be considered for magnetic resonance imaging and/or endoscopic ultrasound immediately. Type 2 diabetes with significant weight loss (>5% of body weight within the last 6 months) should have urgent pancreatic protocol CT imaging. The vast majority of new-onset diabetes mellitus (~95%) should be referred for a standard transabdominal ultrasound first in order to identify and separate a possible subgroup of advanced PDAC [145,146]. The remaining patients should be resolved individually (Figure 1).

Should transabdominal ultrasound be recommended as the first step in the vast majority of new-onset diabetes mellitus cases, such an approach needs to be supported by a feasibility estimation and capacity balance sheet. According to the Institute of Health Information and Statistics of the Czech Republic, about 3,000,000 transabdominal ultrasound investigations are performed in the Czech Republic per year, with a slightly increasing trend within the past five years (from 2,829,511 in 2018 to 3,109,057 in 2022) [147]. Even in a case of full compliance, 30 thousand additional transabdominal ultrasound investigations would represent an increase of only 1%.

Last but not least, it is necessary to mention the irreplaceable role of multidisciplinary teams. All PDAC cases must be evaluated and appraised by a multidisciplinary team (MDT) that can decide the further course, follow-up, and planned therapy.

We are aware of the possible limitations of our concept. A prospective multicenter study is needed to validate the efficacy of this approach. Prompt access to CT imaging (pancreatic protocol) and/or MRI and/or EUS (with fine-needle aspiration/biopsy) might be a limiting factor. Our concept does not cover younger patients and/or subjects without diabetes mellitus or individuals with prediabetes [148].

## 11. Conclusions

A structured, well-differentiated, and elaborately designed approach to older patients (over 60 years of age) with new-onset diabetes mellitus could improve the current unsatisfactory situation in diagnostics and subsequent poor outcomes of treatment of PDAC. Our concept includes such an algorithm. Further studies are warranted.

## Figures and Tables

**Figure 1 cancers-15-03669-f001:**
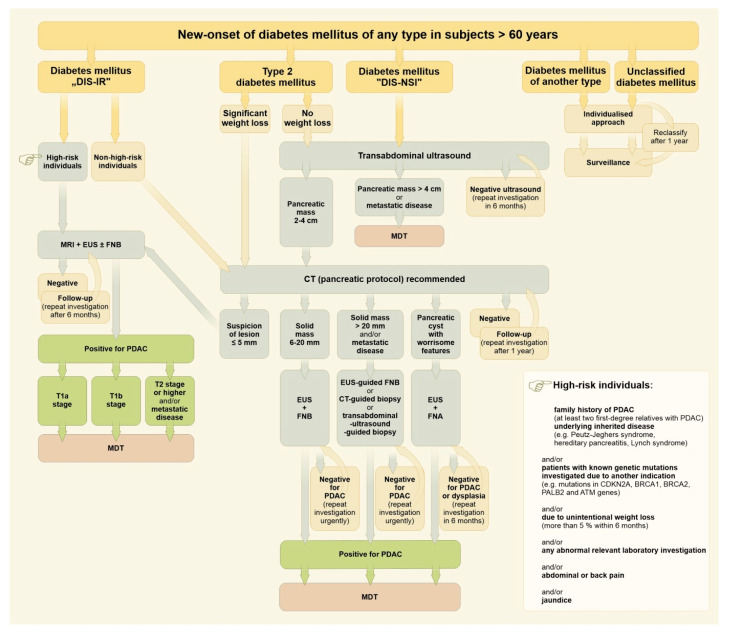
Algorithm for the management of subjects over 60 with new-onset diabetes mellitus of any type. Drawing by Hana Kotlandova. Note: Diabetes mellitus “DIS-IR” in this context means diabetes mellitus characterized by decreased insulin secretion and increased insulin resistance. Diabetes mellitus “DIS-NSI” in this context means diabetes mellitus characterized by decreased insulin secretion and normal sensitivity to insulin.

**Table 1 cancers-15-03669-t001:** Typical basic characteristics of new-onset type 2, PDAC-associated type 3c, and diabetes mellitus associated with non-malignant diseases of the exocrine pancreas. Simplified and partly adopted from References [25,54,84,85,86,87,88,89,90,91,92,93,94,95,96,97,98,99,100,101,102,103,104,105,106,107,108,109,110,111].

Parameter	Type 2 Diabetes Mellitus	PDAC-Associated Type 3c Diabetes Mellitus (Usually Associated with Weight Loss and Early Need for Insulin Therapy)	Diabetes Mellitus Associated with Non-Malignant Diseases of the Exocrine Pancreas
Common average age (years) at a new onset of diabetes mellitus	50–60 (young-onset < 45)	>60	~45
Males/females ratio	2:1	2:1	3:1
Underlying disease (typical examples)	Metabolic syndrome	Pancreatic cancer	Chronic alcohol-induced pancreatitis
Body weight	Stable or increased	Significantly decreased	Stable or decreased
Fasting glycemia (without current therapy for diabetes mellitus)	Increased	Increased	Slightly increased
Glycosylated hemoglobin (HbA1c)	Increased	Increased	Increased
Fasting serum insulin	Increased	Decreased	Usually decreased
C-peptide fasting/postprandial	Not decreased	Decreased	Decreased
Fasting and stimulated serum glucagon	Increased	Increased	Decreased
Fasting and stimulated serum pancreatic polypeptide	Normal	Decreased	Decreased
Insulin resistance	Present	Present	Absent
HOMA-IR index	High	High	Normal
QUICKI index	Low	Low	Normal

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
