# Peer review of "Diabetes Mellitus in Pancreatic Cancer: A Distinct Approach to Older Subjects with New-Onset Diabetes Mellitus"

_cancers, 2023, doi:10.3390/cancers15143669_

Round 1
Reviewer 1 Report
This is a well written review in the first half, but the main point of the article about the algorithm is lacking. There needs to be more explanation, data and rationale for the algorithm proposed.
Side Note: A table summarizing the Surveillance of Pancreatic Cystic Lesions would be great for those who don't understand them.
Lines 205-207, I do not understand the statement “This type of diabetes mellitus is prevalent even in small pancreatic cancers and occurs before the PDAC is radiologically detectable, in contrast to type 2 diabetes mellitus.” It sounds like you are saying type 2 diabetes mellitus is seen in PDAC at a different time?
Line 215 I think you are missing “is” as in “This type of diabetes mellitus is harder to control.”
Lines 271-273 seems to use the same formula regardless of units.
I have several issues with Figure 1 which is in the end, the main point of this paper.
Why do you recommend MRI + EUS +/- FNB for DM DIS-IR but others your recommend an US first. This should be discussed in the body of the text and defended with data / references if able.
For Type 2 DM and DM DIS-NSI it seems the first branch point should be weight loss, yes or no. If no, get an ultrasound. If yes, frankly everyone gets a CT Pancreas Protocol.
From there the small lesion gets an MRI, the solid masses get EUS+FNB. Maybe metastatic disease should still be here with referral to MDT.
Everyone who is positive for PDAC should evaluated by a multidisciplinary team. I think all of your recommendations once PDAC is identified should be MDT and end. The management of T1a, T1b and T2 or higher should not be on this algorithm.
Author Response
We want to thank reviewers for their valuable comments and suggestions. We appreciate reviewers' help very much.
Reviewer 1
|
Thank you for your review. |
|
Comments and Suggestions for Authors
This is a well written review in the first half, but the main point of the article about the algorithm is lacking. There needs to be more explanation, data and rationale for the algorithm proposed.
Side Note: A table summarizing the Surveillance of Pancreatic Cystic Lesions would be great for those who don't understand them.
We did not include any new table as pancreatic cystic lesions are not the main issue. Algorithm of European Guidelines (Gut 2018; 67(5): 789-804) are based on different indications for surgery. Nevertheless, we have added an important information on cysts associated with diabetes mellitus, just to make these recommendations more understandable.
Lines 205-207, I do not understand the statement “This type of diabetes mellitus is prevalent even in small pancreatic cancers and occurs before the PDAC is radiologically detectable, in contrast to type 2 diabetes mellitus.” It sounds like you are saying type 2 diabetes mellitus is seen in PDAC at a different time?
Thank you, you are right. We have edited the text.
Line 215 I think you are missing “is” as in “This type of diabetes mellitus is harder to control.”
Yes, thank you.
Lines 271-273 seems to use the same formula regardless of units.
Yes, thank you. We have edited the text.
I have several issues with Figure 1 which is in the end, the main point of this paper.
Why do you recommend MRI + EUS +/- FNB for DM DIS-IR but others your recommend an US first. This should be discussed in the body of the text and defended with data / references if able.
This is an important point, thanks for it. Most cases of new-onset diabetes mellitus are type 2. It would not be feasible to perform MRI + EUS +/- FNB in all. That is why transabdominal ultrasound is recommended in those patients without decreased insulin secretion and increased insulin resistance and/or without alarm symptoms. We have updated Figure 1 according to your recommendation. We have edited the text and added data on quantity balance and feasibility estimations, supported by exact numbers from the Czech registry. We have also added some more references.
For Type 2 DM and DM DIS-NSI it seems the first branch point should be weight loss, yes or no. If no, get an ultrasound. If yes, frankly everyone gets a CT Pancreas Protocol.
Yes, we agree. We have updated Figure 1 according to your recommendation.
From there the small lesion gets an MRI, the solid masses get EUS+FNB. Maybe metastatic disease should still be here with referral to MDT.
Yes, we agree, we have updated Figure 1 according to your recommendation.
Everyone who is positive for PDAC should evaluated by a multidisciplinary team. I think all of your recommendations once PDAC is identified should be MDT and end. The management of T1a, T1b and T2 or higher should not be on this algorithm.
Yes, we agree, all PDAC cases must be evaluated and appraised by a multi-disciplinary team. Nevertheless, we are convinced that it is relevant to mention a rough therapeutic framework. We have edited the text to make this point of view clear.

Reviewer 2 Report
In this article the authors review the data on diabetes mellitus in pancreatic cancer. Additionally, the authors provide 'the structured, well differentiated and elaborately designed approach to elderly with a new onset of diabetes mellitus' that 'could improve the poor current situation in diagnostics and subsequent therapy of PDAC.'
General comment:
The topic of the review is relevant. The article is well written and it clarifies the terms related to different types of diabetes mellitus and pancreatic cancer in a concise and informative way. The suggested concept might prove relevant in the diagnostics and therapy of PDAC. The limitations of the proposed concept are fairly presented.
Specific point:
Title could mention the new approach to elderly with a new onset of diabetes mellitus that the authors are suggesting. This is only a suggestion and I leave it up to the authors to decide on its suitability.
Minor points:
Figure 1: The font size could be increased.
Line 52: The authors could mention the date when PubMed database was screened for publications.
Author Response
Reviewer 2
|
Thank you for your review. |
Začátek formuláře
In this article the authors review the data on diabetes mellitus in pancreatic cancer. Additionally, the authors provide 'the structured, well differentiated and elaborately designed approach to elderly with a new onset of diabetes mellitus' that 'could improve the poor current situation in diagnostics and subsequent therapy of PDAC.'
General comment:
The topic of the review is relevant. The article is well written and it clarifies the terms related to different types of diabetes mellitus and pancreatic cancer in a concise and informative way. The suggested concept might prove relevant in the diagnostics and therapy of PDAC. The limitations of the proposed concept are fairly presented.
Specific point:
Title could mention the new approach to elderly with a new onset of diabetes mellitus that the authors are suggesting. This is only a suggestion and I leave it up to the authors to decide on its suitability.
That you for your kind suggestion. We want to stay humble, as we do not know yet if our proposal would work, would be feasible and thus would become as a real new approach.
Minor points:
Figure 1: The font size could be increased.
Font size is limited by the size of page and figure resolution. That is why we did increase the font size.
Line 52: The authors could mention the date when PubMed database was screened for publications.
Thank you, we have added it.
Round 2
Reviewer 1 Report
The version available to me still has the wrong HOMA-IR formula.
I believe it is Insulin (mU/L) * Glucose (mg/dL)/400
The main focus of this article is the diagnosis, hopefully early, of patients with pancreas cancer based on being elderly (above age 60) and having diabetes.
It does not seem you are in any position to flow the expected treatment after MDT, so I still have issue with the implication of what to do with T1a, T1b and T2 or higher lesion.
Practically the Transabdominal US if positive ends in "staging" which "CT" seems to be the same thing.
I still think this figure could be simplified and would avoid recommendation of treatment of pancreas cancer, which is out of scope.
Author Response
Reviewer 2
Thank you for your comments.
The version available to me still has the wrong HOMA-IR formula.
I believe it is Insulin (mU/L) * Glucose (mg/dL)/400
Thank you, the formula was corrected.
The main focus of this article is the diagnosis, hopefully early, of patients with pancreas cancer based on being elderly (above age 60) and having diabetes.
It does not seem you are in any position to flow the expected treatment after MDT, so I still have issue with the implication of what to do with T1a, T1b and T2 or higher lesion.
We accept your recommendation and have deleted all notes on possible therapy.
Practically the Transabdominal US if positive ends in "staging" which "CT" seems to be the same thing.
Thank you. We have simplified this part of the Figure. Advanced pancreatic cancer and/or metastatic disease diagnosed by transabdominal ultrasound should be evaluated by MDT. CT is not necessarily the same thing, just one of possible options.
I still think this figure could be simplified and would avoid recommendation of treatment of pancreas cancer, which is out of scope.
We accept your recommendation and have deleted all notes on possible therapy.

Round 3
Reviewer 1 Report
I think the flow chart could still be further improved, but it is acceptable as is. I hope this paper helps move people to earlier diagnosis.